# Proteomics reveals differentially regulated pathways when comparing grade 2 and 4 astrocytomas

Denildo C. A. Verissimo[1,2☯], Amanda C. Camillo-Andrade[1☯], Marlon D. M. Santos[1], Sergio L. Sprengel[2], Simone C. Zanine[2], Luis A. B. Borba[2], Paulo C. Carvalho[1]*, Juliana de S. da G. Fischer [1]*

1 Laboratory for Structural and Computational Proteomics—Carlos Chagas Institute, Fiocruz Paraná, Curitiba, PR, Brazil, 2 Clinical Hospital of the Federal University of Paraná, Curitiba, Paraná, Brazil

☯ These authors contributed equally to this work.
* paulo@pcarvalho.com (PCC); julifr@gmail.com (JSGF)

**Data Availability Statement:** The mass spectrometry data was deposited to the ProteomeXchange Consortium via the PRIDE (83)

## Abstract

Astrocytic tumors are known for their high progression capacity and high mortality rates; in this regard, proteins correlated to prognosis can aid medical conduct. Although several genetic changes related to progression from grade 2 to grade 4 astrocytoma are already known, mRNA copies do not necessarily correlate with protein abundance and therefore could shadow further comprehension about this tumor's biology. This motivates us to seek for complementary strategies to study tumor progression at the protein level. Here we compare the proteomic profile of biopsies from patients with grade 2 (diffuse, $n = 6$) versus grade 4 astrocytomas (glioblastomas, $n = 10$) using shotgun proteomics. Data analysis performed with PatternLab for proteomics identified 5,206 and 6,004 proteins in the 2- and 4-grade groups, respectively. Our results revealed seventy-four differentially abundant proteins ($p < 0.01$); we then shortlist those related to greater malignancy. We also describe molecular pathways distinctly activated in the two groups, such as differences in the organization of the extracellular matrix, decisive both in tumor invasiveness and in signaling for cell division, which, together with marked contrasts in energy metabolism, are determining factors in the speed of growth and dissemination of these neoplasms. The degradation pathways of GABA, enriched in the grade 2 group, is consistent with a favorable prognosis. Other functions such as platelet degranulation, apoptosis, and activation of the MAPK pathway were correlated to grade 4 tumors and, consequently, unfavorable prognoses. Our results provide an important survey of molecular pathways involved in glioma pathogenesis for these histopathological groups.

## Introduction

Gliomas are the most common primary brain tumors, accounting for more than 70% of all primary central nervous system (CNS) neoplasms. They are known for their high progression

partner repository with the dataset identifier PXD033782.

**Funding:** JSGF received a scholarship from covenant APU / PUCPR 28/2021 (Fundação Araucária); Carlos Chagas Institute (Fiocruz Paraná) for reagents and infrastructure support. We also acknowledge the grant PEP-ICC-008-FIO-21 from Fiocruz. The funders had no role in study design, data collection and analysis, decision to publish, or preparation of the manuscript.

**Competing interests:** The authors have declared that no competing interests exist.

capacity and for leading to increased mortality rates. Astrocytic tumors are neuroepithelial tumors originating from the supporting glial cells of the CNS [1]. They can occur from the first decade of life and manifest in any location of the central nervous system (CNS) [2].

The worldwide incidence of tumors of the nervous system in 2020 was about 300,000 new cases (17th position among registered types of cancer) and presented mortality of 241,037 as per Word Health Organization (WHO) (12th position) [3]. In Brazil, according to Cancer National Institute (INCA) data, the estimated incidence for 2022 is more than 11,000 new cases, and the mortality in 2020 was 9,309, following the relatively high global rates.

Proteomics was previously used by Gollapalli *et* al. to analyze various grades of glioma tissues. The authors pinpointed several metabolic pathways such as glycolysis, TCA-cycle, electron transport chain, lactate metabolism, and blood coagulation pathways that were altered in gliomas. Most of the more abundant proteins in gliomas were correlated with redox reactions, protein folding, pre-messenger RNA (mRNA) processing, antiapoptotic, and blood coagulation [4].

Although several genetic changes related to progression from grade 2 to grade 4 astrocytoma are already known [5], mRNA copies do not necessarily correlate with protein abundance. Here we use shotgun proteomics to shortlist pathways altered during the progression of astrocytes tumors. To achieve this, we compared proteomic profiles of biopsies from patients with grade 2- (diffuse, $n = 6$) versus grade 4 astrocytoma (glioblastomas, $n = 10$). Our results provide an important survey of molecular pathways involved in astrocytoma pathogenesis for these histopathological groups.

## Materials and methods

### Materials

Qubit® Protein Assay Kit (cat. no. Q33212) and RapiGest™ SF acid-labile surfactant (cat. no. 186001861) were acquired from Invitrogen and Waters, respectively. Sequence grade modified trypsin (V511A) was purchased from Promega. All other laboratory reagents were bought from Sigma-Aldrich (St. Louis, MO), unless stated otherwise.

### Methods

**Patients and ethics aspects.** All methods were performed in accordance with the relevant guidelines and regulations of Brazil. This study was approved by the Ethics Committee of Oswaldo Cruz Foundation and the Federal University of Clinical Hospital of Parana under the numbers CAAE: 63056316.8.0000.5248 and 63056316.8.3001.0096, respectively. Informed written consent was obtained from all subjects and/or their legal guardian(s) before participating in the study. All participants were provided with detailed information about the study, its objectives, and the use of their data for research purposes. No minors participated in this study. All data were fully anonymized before analysis to ensure participant privacy.

After being collected by a neurosurgeon belonging to the staff of the Clinical Hospital of the Federal University of Paraná, the tumor samples were immediately put on dry ice and then stored in a -80˚C freezer until used. This study includes sixteen patients with astrocytoma; six diagnosed with grade 2 and ten with grade 4, respectively. The characteristics of the patients included in this study are presented in Table 1.

All astrocytomas grade 2 and all grade 4 were classified as NOS (Not Otherwise Specified) according to histopathological features and patient's history in accordance with WHO CNS5 specifications.

**Sample preparation.** The set of sixteen tissue samples was pulverized under liquid nitrogen, following a previously described technique [6]. Protein extraction was then carried out

**Table 1. Characteristics of the patients included in this study, indicating the age, gender, and histopathological diagnosis.**

| Patient ID | Age | Gender | Tumor grade |
|---|---|---|---|
| 11 | 19 | Male | 2 |
| 31 | 34 | Female | 2 |
| 48 | 18 | Male | 2 |
| 95 | 18 | Male | 2 |
| 106 | 27 | Female | 2 |
| 113 | 58 | Male | 2 |
| 25 | 75 | Female | 4 |
| 26 | 41 | Male | 4 |
| 59 | 64 | Male | 4 |
| 39 | 55 | Male | 4 |
| 53 | 72 | Female | 4 |
| 69 | 77 | Female | 4 |
| 46 | 47 | Female | 4 |
| 108 | 50 | Male | 4 |
| 112 | 74 | Male | 4 |
| 29 | 57 | Female | 4 |

using a solution of 0.1% RapiGest (w/v) in 50 mM triethylammonium bicarbonate (TEAB). Subsequently, centrifugation at 18,000 x g at 4˚C for 15 minutes took place, and protein quantification was performed with the Qubit 3.0 fluorometric assay in accordance with the manufacturer's protocol. A total of 100 μg protein from each sample was subjected to reduction with 10 mM dithiothreitol (DTT) at 60˚C for 30 minutes. After cooling the samples to room temperature, they were incubated in the absence of light with 25 mM iodoacetamide (IAA) for 30 minutes. Lastly, the samples were digested over 20 hours with high sequence grade modified trypsin, utilizing a 1:50 (Enzyme/Substrate) ratio at 37˚C [7].

**Desalting and sample quantification.** The enzymatic reaction was terminated by adding trifluoroacetic acid (0.4% v/v final) to the samples, and peptides were incubated for an additional 45 minutes to decompose RapiGest. The samples were then centrifuged at 18,000 x g for 15 minutes to remove any remaining insoluble components. Following this step, peptide quantification was performed using the Qubit 3.0 (Invitrogen) fluorometric assay, as per the manufacturer's guidelines. Each sample was desalted and concentrated employing Stage-Tips (STop and Go-Extraction TIPs), based on the method established by Rappsilber et al. [8].

**Mass spectrometry acquisition.** Peptides were analyzed via LC-MS/MS (Liquid chromatography with tandem mass spectrometry) using an UltiMate 3000 (Thermo Fisher®) ultra-high-performance liquid chromatography (UHPLC) system connected to an Orbitrap FusionTM LumosTM mass spectrometer (Thermo, San José). Peptide mixtures were loaded onto a 75 mm i.d., 30 cm long column packed in-house with 3 μm ReproSil-Pur C18-AQ resin (Dr. Maisch) at a flow rate of 250 nL/min and subsequently eluted at a flow rate of 250 nL/min from 5% to 40% ACN in 0.1% formic acid over a 140 min gradient [9]. The mass spectrometer was configured in data-dependent acquisition (DDA) mode, allowing automatic switching between full-scan (MS) and MS/MS (MS2) acquisition. Survey MS spectra (from m/z 300–1,500) were captured in the Orbitrap analyzer at a resolution of 120,000 at m/z 200. The most intense ions obtained within a 2s cycle time were selected, excluding unassigned or 1+ charge state ions. These ions were sequentially isolated and fragmented using higher-energy collisional dissociation (HCD) with a normalized energy of 30. Fragment ions were analyzed at a

resolution of 15,000 at 200 $m/z$. General mass spectrometric conditions included a spray voltage of 2.5 kV, no sheath or auxiliary gas flow, ion transfer tube temperature at 250°C, predictive automatic gain control (AGC) enabled, and an S-lens RF level of 40%. The mass spectrometer scan functions and nLC solvent gradients were controlled using the Xcalibur 4.1 data system (Thermo, San José). Each biological replicate underwent two technical replicates.

**Peptide spectrum matching (PSM).** Data analysis was conducted using the PatternLab for proteomics software, which is freely accessible at http://www.patternlabforproteomics.org [10, 11]. Homo sapiens sequences were retrieved on June 6th, 2020, from Swiss-Prot and subsequently utilized to generate a target-decoy database, which included a reversed version of each sequence along with 104 common mass spectrometry contaminants. The Comet 2016.01 rev. 3 search engine was employed to identify the mass spectra [12].

The search parameters took into account fully and semi-tryptic peptide candidates with masses ranging from 550 to 5,500 Da, up to two missed cleavages, 40 ppm for precursor mass, and bins of 0.02 m/z for MS/MS. Fixed and variable modifications considered were carbamidomethylation of cysteine and oxidation of methionine, respectively.

**Validation of peptide-spectrum match.** Evaluation of PSM validity was performed using the Search Engine Processor (SEPro) [13]. The identifications were organized by charge state (2+ and ≥ 3+) and tryptic status, generating four distinct subgroups. A Bayesian discriminator was created for each group using XCorr, DeltaCN, DeltaPPM, and Peaks Matches values. The identifications were sorted in a non-decreasing order based on the discriminator score. A cut-off score allowed a false-discovery rate (FDR) of 2% at the peptide level, determined by the number of decoys [14].

This procedure was executed independently for each data subset, resulting in an FDR not influenced by charge state or tryptic status. Moreover, a minimum sequence length of five amino acid residues and a protein score exceeding three were imposed. Lastly, identifications diverging by over 10 ppm from the average were excluded. The implementation of these postprocessing filters led to FDRs at the protein level being less than 1% for all search outcomes [15].

**Relative protein quantification.** The quantification was carried out according to the PatternLab's Normalized Ion Abundance Factors (NIAF) as a relative quantification strategy. We remind that the NIAF is equivalent to the NSAF [16] but applied to the extracted ion chromatogram (XIC) [17].

Quantification was conducted using PatternLab's Normalized Ion Abundance Factors (NIAF) as a strategy for relative quantification. It is worth noting that NIAF is analogous to the NSAF [16] but applied to the extracted ion chromatogram (XIC) [17].

**Pinpointing differentially abundant proteins and data analysis.** We employed PatternLab's TFold module [18] to pinpoint proteins exhibiting differential abundance between grades 2 and 4 groups. Our proteomic comparison considered only proteins identified with a minimum of two unique peptides (peptides mapping to a single database sequence) and a q-value ≤ 0.1. Proteins present in at least two biological replicates were considered for TFold analysis. This method is concisely outlined in our bioinformatics protocol [10, 11]. Protein pathways were mapped using Reactome [19].

## Results and discussion

### Protein identification in grade 2 and grade 4 astrocytomas

Here we use shotgun proteomics to shortlist alterations in the protein pathways when comparing biopsies of grade 2 versus grade 4 astrocytomas. The higher number of grade 4 astrocytoma (ten) is in line with its greater prevalence in the population [20].

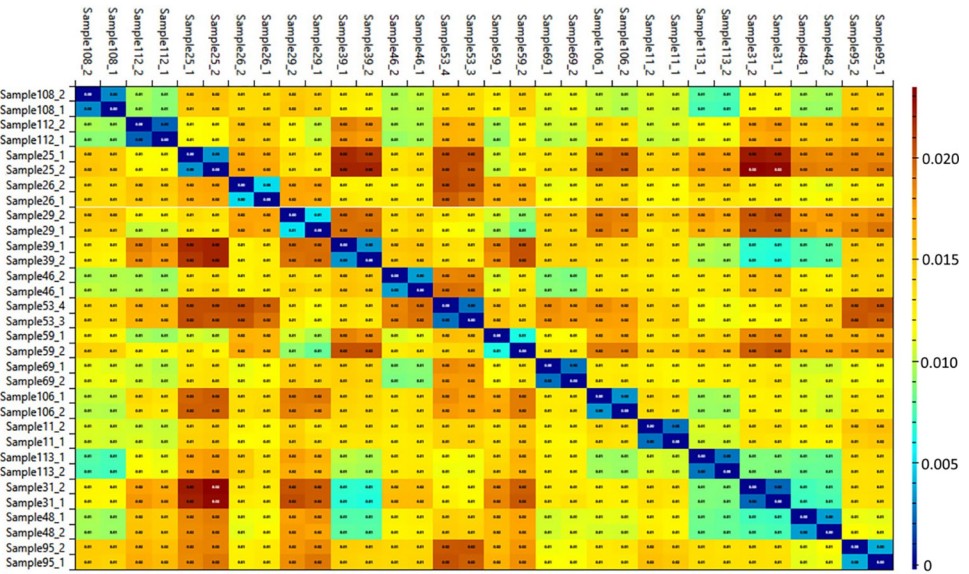

**Fig 1. Heatmap comparing grade 2 and 4 astrocytomas studied.** The colors represent the similarity between samples. The closer to zero (blue), the greater the similarity.

For the group of grade 2 astrocytomas, we identified 52,133 peptides mapping to 5,816 proteins (5,206 with maximum parsimony). In the grade 4 group, we identified 62,759 peptides mapping to 6,686 proteins (6,004 with maximum parsimony).

We evaluated the reproducibility of our technical replicates with the RawVegetable program [21]; a heatmap is presented in Fig 1.

Our heatmap's lower-right corner presents a greater color uniformity, indirectly reflecting less proteomic variability among the grade 2 tumors.

We used PatternLab's Venn diagram module to list proteins uniquely identified in each group of tumors studied (Fig 2).

The Ven Diagram shows that 748 proteins were identified only in the grade 4 astrocytoma, while 174 proteins were exclusively for the grade 2 astrocytoma. As both tumor types originate from the same cell type, most proteins are common (2,658 proteins). The complete list of proteins found in each condition is available in S1 Table.

### Differentially abundant proteins between grade 2 and grade 4 astrocytomas

The differentially abundant proteins between grade 2 and 4 were analyzed with the TFold module of PatternLab for Proteomics. The abundance of each protein was statistically evaluated under the criteria of a variable fold change and satisfying a $q$-value (Fig 3). The seventy-four differential abundant proteins list is available in S2 Table.

We used the Reactome (reactome.org) platform [19] to list the enriched metabolic pathways, for each grade tumor, under the light of differentially abundant proteins (blue dots in Fig 3) (Table 2).

### Proteins exclusively identified in the grade 4 astrocytomas

Among the proteins identified only in the grade 4 astrocytomas, we highlight Rho GTPase-activating proteins, Cathepsin and S-100.

The Rho GTPase family is involved in critical cellular functions such as cell polarity, vesicular traffic, cell cycle, and transcriptome dynamics [22]. Due to this relationship with cancer,

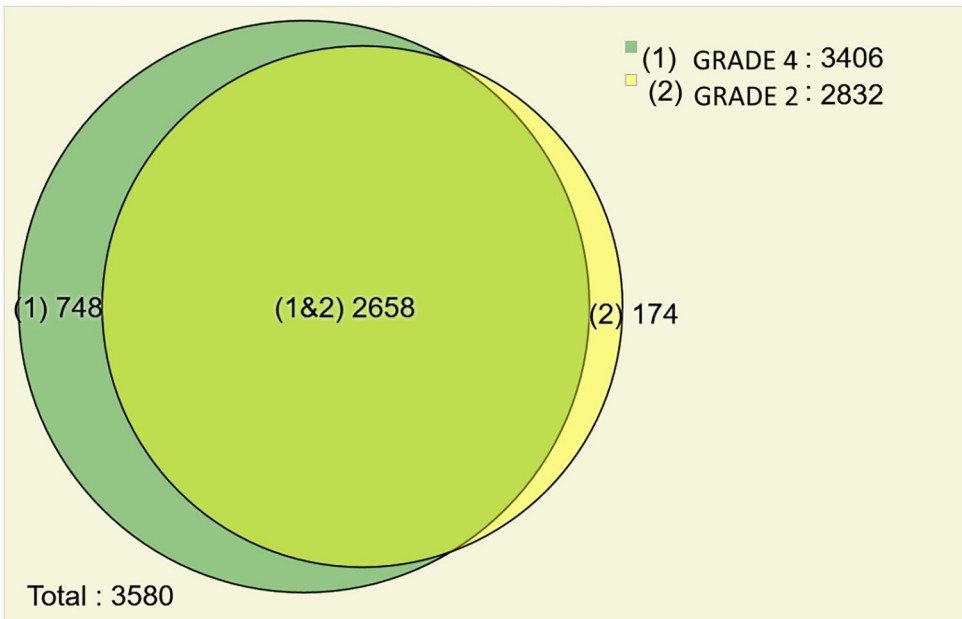

**Fig 2. Venn diagram of proteins identified in the grade 2 and 4 astrocytomas.** We pinpointed 748 and 174 proteins exclusive of the grade 4 and grade 2 astrocytomas, respectively.

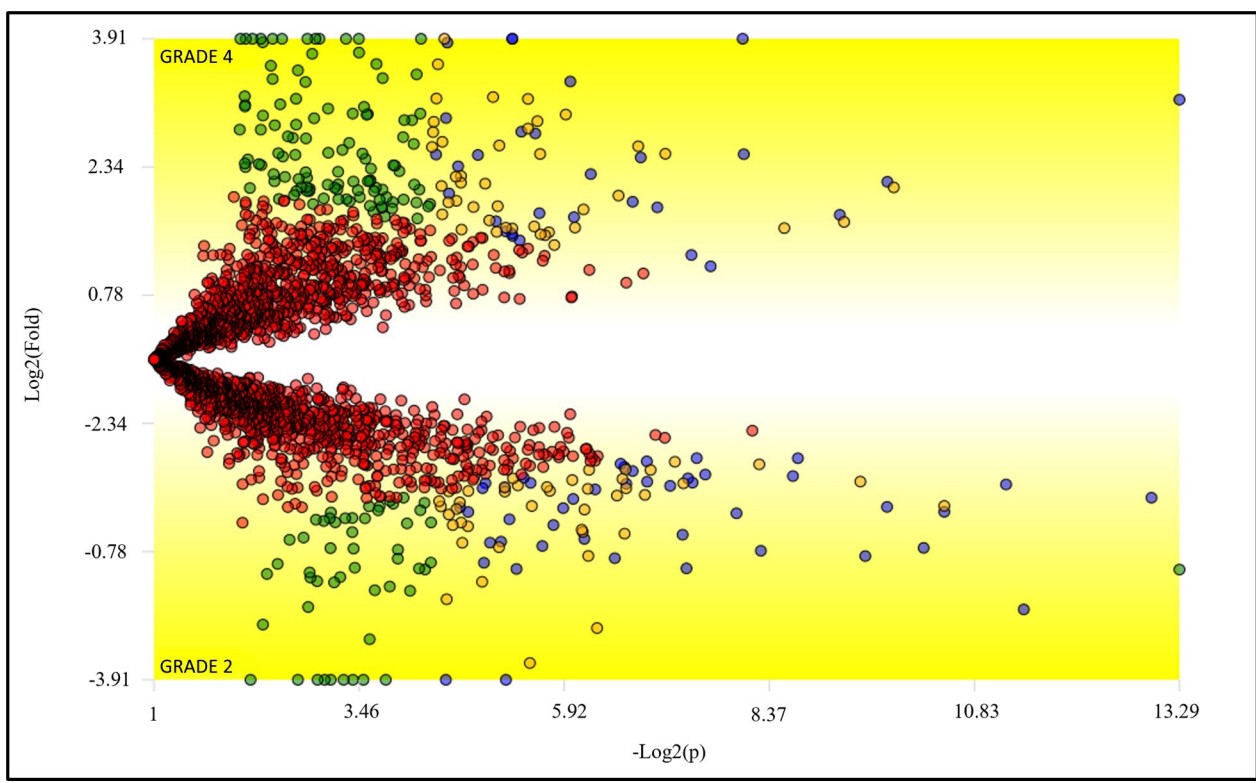

**Fig 3. Differentially abundant proteins between grade 2 and grade 4 astrocytomas according to PatternLab for proteomics TFold module.** Each "dot" represents a protein. The blue dots represent the 74 proteins that meet both statistical analyses plus the Benjamini Hochberg FDR correction between the two groups and are differential abundant. Orange dots indicate the 84 proteins that passed the t-test criterion but not the fold change criterion. The red dots represent not differentially abundant proteins in either of the two criteria. The green dots indicate the 165 proteins that have passed the fold change criterion. However, the t-test indicates that its differential abundance was random.

**Table 2. Enriched metabolic pathways for grades 2 and 4 astrocytomas.**

| Grade 2 | Grade 4 |
|---|---|
| ECM proteoglycans | Integrin cell surface interactions |
| Pyruvate metabolism and Citric Acid (TCA) cycle | Fibronectin matrix formation |
| Mitochondrial protein import | Peroxisomal protein import |
| Metabolism of carbohydrates | Oncogenic MAPK signaling |
| Metabolism of lipids | Platelet Degranulation |
| Degradation of GABA | Integrin Signaling |
| Diseases associated with glycosaminoglycan metabolism | Programmed cell death and apoptotic execution phase |
| Axon guidance–NCAM1 signaling | Axon guidance–L1CAM interactions |

MAPK–Mitogen Activated Protein kinase; GABA—Gamma-Amino Butyric Acid.

this pathway may be an important therapeutic target [22]. Kwiatkowska *et* al. silenced RhoG using siRNA and showed marginal effects on GBM cell proliferation, but it significantly inhibited cell survival in colony formation assays [23].

According to Fortin Ensign *et* al., the study of aberrant Rho GTPase signaling in grade 4 astrocytoma such as glioblastoma is thus an essential investigation of cell invasion as well as treatment resistance and disease progression [24]. Our results corroborate with the literature and further suggest its potential therapeutic use.

Another protein exclusively identified in grade 4 astrocytoma is cathepsin. Cathepsins are cysteine proteases present in lysosomes and activated at acid pH. These proteins were shown to attenuate the antitumor immune response and act in the degradation of the extracellular matrix, cell invasion, and stimulation of mitosis [25]. This class of proteins was identified in glioblastomas, representing important activity in tumor progression and invasion, besides being indicated as possible therapeutic targets [26].

S100 proteins comprise a group of 21 proteins involved in several intra- and extracellular actions such as calcium homeostasis, regulation of protein phosphorylation, transcription factors, cytoskeleton component dynamics, inflammatory response, growth, and cell differentiation [27]. The biology of this family is complex and multifunctional, contributing to tumorigenesis through its mediation in phenomena such as cell proliferation, metastasis, angiogenesis, and immune evasion, and some protein subtypes are already considered a therapeutic target of drugs in clinical trials. [28]. Lu *et* al. demonstrated that S100 A12 was upregulated in tissues of glioma patients, and the expression was correlated to WHO stage and tumor size. Besides, the authors showed that knockdown of S100 A12 inhibits the proliferation, migration, and invasion of glioma cells by regulating cell apoptosis and EMT [29].

## Proteins identified exclusively in the grade 2 astrocytomas

Among the proteins identified only in the grade 2 astrocytoma we highlight the Heat shock protein β-8, Long-chain-fatty-acid—CoA ligase, and Voltage-gated potassium channel.

Identified initially as heat stress-responsive proteins, acting as molecular chaperones (helper proteins), heat shock proteins (HSPs) have the role of maintaining intracellular functions in response to a wide variety of perturbations (physical or chemical) or even under alterations in physiological conditions, preventing cell death [30]. These proteins have been reported to be more abundant in cancer as a response to internal stress developed by the tumor cells. Some examples of stress causes are the breakdown of exacerbated protein synthesis or the presence of mutant proteins and hypoxia, nutrient deprivation, and acidosis [31].

Thus, in tumors, HSPs promote resistance to apoptosis, activation of the cell stroma, inducing progression, angiogenesis, and tissue invasiveness [32]. Research on these proteins is still preliminary in gliomas. HSPs have also been correlated to survival time and prognosis for both good and bad [33].

Fatty acids are essential nutrients and function as essential building blocks of the body and participate in energy metabolism and cellular signaling mechanisms. However, given the growing number of overweight and obese individuals, excess fatty acid metabolism has been increasingly associated with metabolic disorders and carcinogenesis [34]. The overexpression of Long Chain Fatty Acid Ligase (ACSL) has been reported in several types of cancers, such as human hepatocellular carcinoma, colon adenocarcinoma [35], and breast cancer, inferring a role in the carcinogenesis of these neoplasms [36]. In gliomas, this molecule was related to cell survival, especially in extracellular acidic environments [37], which is compatible with our results, being more frequent in grade 2 tumors, with better prognosis.

## Differential abundant proteins in the grade 2 group

We used Reactome platform to help understand the role of the differential abundant proteins in each group. We found proteoglycans pathway, composed of hyaluronan and proteoglycan link protein, versican core protein, brevican core protein, and tenascin-R to be differential abundant in the grade 2 astrocytomas.

Hyaluronic acid (HA) is one of the main components of the brain extracellular matrix (ECM), constituting a three-dimensional extracellular mesh by connecting to binding proteins and proteoglycans [38, 39]. In tumors, its concentration is increased and acts as a stimulator for cell proliferation, mobilization, and invasion. In addition to inducing the production of numerous factors related to migration, such as the CD44 antigen, PTEN (phosphatase and tensin homolog), and matrix metalloproteinases, HA facilitates the spatial expansion of the tumor and assists in cell adhesion to occupied spaces through interactions with their binding molecules [40].

The HA-binding proteins and proteoglycans increased in the grade 2 samples could stabilize the proteoglycan aggregates from their natural dissociation, and protected their degradation under conditions of tissue catabolism activation [41].

Versican protein, along with Brevican protein, tenascin-R, AH-binding proteins, and the proteoglycans discussed above, are all more abundant in the grade 2 samples and constitute the basis of the HA-related complex commonly present in the ECM of mature brains [40, 42]. The Brevican protein, after its proteolysis by ADAMTS4, binds to fibronectin in cells leading to the production of integrins, inducing tumor motility and invasion [43, 44]. In addition to participating in ECM formation, tenascin-R and Brevican protein also functions in the modulation of regulatory and inhibitory synapses [45].

In agreement with our results, tenascin-R and other proteins of the tenascin family are more closely related to grade 2, showing decreasing levels of concentration in grade 4 gliomas [46, 47].

Regarding cell adhesion molecules, NCAM pathways were differentially activated in the grade 2 group through the overexpression of NCAM1 and PRN (Major Prion Protein). Researches published in the international literature indicate that the negative regulation of NCAM may contribute to the invasion of gliomas, promoting cell breakdown and secretion of proteases [48, 49], and their positive regulation (hyperexpression) is one of the determining factors for better prognosis of grade 2 tumors. Furthermore, the increased expression of this protein opens the possibility of a possible target of grade 2 to immunotherapies anti-NCAM [50].

We also identified a predominance of expression of citric acid cycle proteins (Krebs cycle), including isocitrate dehydrogenase type 2 (IDH-2), and oxidative phosphorylation, including NADP, in the grade 2 tumors, a fact also reinforced by the over-expression of mitochondrial import protein pathway (ADP/ATP translocase 1).

Besides glucose, other sources of acetyl-CoA for the Krebs Cycle include the breakdown of free fatty acids and ketone bodies in a fasting state. Based on this information, Schwartz *et al.* advocated in favor of a ketogenic diet for patients with gliomas [51]. That way induces increased mitochondrial metabolism in these tumors and potentially increases survival. Complementary clinical studies are still necessary to determine this approach's therapeutic efficacy.

We also observed a pathway related to the degradation of gamma-amino butyric acid (GABA), which is metabolized in the mitochondrial matrix to succinate by the action series of two enzymes, 4-aminobutyrate aminotransferase and semialdehyde succinate dehydrogenase (SSADH) [52], proteins differentially abundant identified in the grade 2 group of our work.

Decreased levels of GABA are associated with increased levels of its metabolic by-product 4-hydroxybutyrate (GHB). This by-product induces GBM stem cells' differentiation and infiltrates deep glioma in less aggressive cells [53]. As such, it can be hypothesized that the intensification of GABA degradation is closely related to the lower aggressiveness of gliomas, as demonstrated in the findings of our work and the increase in the incidence of epilepsy in this class of tumors.

## Abundant proteins in the grade 4 group

The L1CAM pathway, related to cell adhesion molecules, was differentially activated in the grade 4 group through the overexpression of moesin and tubulin. Such hyperactivation is related to a worse prognosis of gliomas as it favors cell proliferation and motility [54, 55]; this is consistent with the behavior of glioblastomas. The inhibition of L1CAM through targeted therapies has shown promising results in inhibiting proliferation and cell invasion by suppressing the growth of gliomas [56, 57].

The fibronectin and fibrinogen gamma chain was also found as spotlights in cell surface integrins interactions and fibronectin matrix formation. Concerning gliomas, as early as 1985, increased levels of fibronectin were detected in the plasma of patients with glioblastoma [58]. In 1997, Ohnishi *et al.* demonstrated the promotion of fibronectin in glioma cell invasion and migration through chemotactic activity [59]. Over the years, the relationship between fibronectin activity and mechanisms of induction of malignancy in gliomas has become even more solid, especially in cell adhesion, differentiation, proliferation, and chemoresistance, as well as their possible mechanisms. Reductions in sox-2 and nestin levels and increased levels of glial fibrillary acidic protein (GFAP) and β-tubulin were also found in glioma lineages, indicating fibronectin-driven cell differentiation.

Additional studies demonstrated that fibronectin facilitated cell proliferation, as evidenced by increased Ki-67 levels, along with the activation of phosphorylated ERK1/2 and cyclin D1. Furthermore, fibronectin was found to inhibit apoptosis mediated by the p53 pathway and enhance the expression of P-glycoprotein [60]. More recent works also evidence this action by activating the PI3K / AKT signaling pathway [61].

In 2019, Golanov *et al.* used brain tissue samples taken during malignant glioma resection surgeries and showed that astrocytes and neurons could produce fibrinogens, albeit only produced in the presence of blood-brain barrier breakdown [62]. In addition to the fibrinogen gamma chain's role in the hemostasis mechanism, it also interacts with TLR-4 (Toll-Like Receptor 4), which is important in innate immunity and neuroinflammation are typical of grade 4 tumors [63, 64]. The greater presence of fibrinogen chains in grade 4 gliomas was also corroborated in another important work on proteomic analysis in different grades of glioma,

showing a positive correlation with increasing grade of gliomas, thus offering insight into the molecular basis behind its aggressive nature [4].

Our results also showed an increased abundance of Ras GTPase activating proteins (GAPs), as also found by Ren et al. [65] in the study of glioma proteomics. As mentioned earlier, this class of proteins is responsible for the inactivation of Ras. Considering the extensive literature demonstrating the hyperactivation of the Ras pathway in gliomas [66–70], our findings of the hyperactivation of its inactivation pathway make us hypothesize a mere biochemical compensation or even a failure of this inactivation mechanism, stimulating the hyperactivation of the genes of Ras GTPases.

Our results showed a significant increase in cytoplasmic isocitrate dehydrogenase type 1 (IDH-1); this has been associated with an overexpression of the non-specific lipid transfer protein, which are members of the complex peroxisomal import protein. These findings support the predominance of cytoplasmic metabolism in grade 4 tumors [71, 72].

Concerning the mechanisms of cell death, in addition to the overexpression of lamin-B1 and BAX protein, denoting a clear performance of apoptosis in grade 4 tumors, we found a differential abundance of Vimentin, recently cited as part of an inhibitory mechanism and responsible for the change in survival achieved in the treatment of cancer patients [73]. We also identified a statistically significant peak of histone H1.5 in this same group. Caspase-mediated changes in this histone family are considered a marker for early apoptosis [74]. Our findings are consistent with those previously reported in the literature [65].

We also found the hyperactivation of platelet degranulation pathways differentially present in the grade 4 tumors group by identifying statistically significant hyperexpression of the H4 heavy chain of inter-alpha-trypsin inhibitor, filamin-A, transgelin-2, and calumenin; these findings are consistent with previously published literature [65].

Platelets possess multifaceted roles in cancer progression, including the ability to safeguard circulating tumor cells from immune surveillance, fostering an environment that supports cell survival. Additionally, they can stimulate invasive characteristics in tumor cells, enhancing their metastatic potential. Platelets also facilitate the transfer of adhesive molecules, which play a crucial part in establishing interactions between tumor cells and the endothelial lining of blood vessels. This interaction aids in the formation of early metastatic niches, further driving the spread of cancer throughout the body [75]. These interactions favor the spread of these cells and cancer-associated thrombogenesis [76].

In 2007, Brockmann *et al.* reported preoperative thrombocytosis as an independent risk factor associated with shorter survival in patients with glioblastoma [77] caused by the growth and development of this tumor [78].

In addition to the relationship with platelet degranulation pathways, calumenin is present in biological functions such as cell adhesion and extracellular matrix organization, all of which are strongly correlated with the epithelial-mesenchymal transition. Its overexpression was related to more aggressive glioma phenotypes (such as wild-type HDI), with worse prognosis, and maybe a strong candidate for a target molecule, depending on future studies [79].

In view of all this, some studies have pointed to a possible antitumor effect of the use of antiplatelet agents, such as aspirin, clopidogrel, and ticagrelor [80, 81], of 10-protein-rich platelets induced by interferon-gamma [82] and the use of genetic engineering promoting platelet production as tumor necrosis factor-related apoptosis-inducing ligand (TRAIL) vectors [83].

## Final considerations

Considering the results taken together, the identified alterations in extracellular matrix organization, cell division signaling, and energy metabolism offer valuable insights into the

mechanisms contributing to the distinct characteristics of these histopathological groups. For example, our study highlights the importance of GABA degradation pathways in grade 2 astrocytomas and their implications for understanding neurotransmitter signaling in glioma pathogenesis. Investigating the relationship between these pathways and tumor cell behavior may help elucidate why grade 2 astrocytomas are associated with a better prognosis. In contrast, the increased presence of platelet degranulation, apoptosis, and MAPK pathway activation in grade 4 astrocytomas suggests a more aggressive tumor phenotype. Understanding how these processes influence tumor growth and invasion could facilitate the identification of therapeutic targets or intervention strategies for grade 4 astrocytomas.

Our study also emphasized the significance of dysregulated energy metabolism in astrocytoma pathogenesis by pinpointing differences in energy metabolism pathways between grades 2 and 4 astrocytomas, which may contribute to their distinct growth rates and invasiveness. Investigating the specific metabolic alterations and their impact on tumor cell proliferation, survival, and migration could reveal novel therapeutic targets to hinder tumor progression.

In all, exploring the interplay between differentially abundant proteins and molecular pathways may provide a more holistic understanding of the complex cellular processes driving astrocytoma development and progression. Further studies examining these interactions to shortlist regulatory nodes are key to better therapeutic targeting to disrupt tumor growth and spread. Future studies should focus on these relationships and the underlying mechanisms, paving the way for more effective and personalized treatment strategies for patients with astrocytomas.

In our study, we acknowledge the limitation of not incorporating all WHO CNS5 guidelines for astrocytoma classification due to the unavailability of certain molecular marker tests, such as IDH mutation status and 1p/19q codeletion, in vast majority of Brazilian public services. This fact is also foreseen in the 5 who classification with the suffix NOS (not otherwise specified). Despite these limitations, our research provides valuable information on the proteomic differences between grade 2 and grade 4 astrocytomas based on their histopathological features and patients' natural history. We recognize that the integration of molecular markers in astrocytoma classification could offer a more accurate diagnosis and improved understanding of tumor biology, leading to better patient stratification and tailored treatment approaches. We encourage future research to build upon our findings and incorporate the WHO CNS5 guidelines for a more comprehensive analysis of astrocytomas and their underlying molecular pathways.

## Conclusion

This work contributes to data on the differentiation of proteomes from grades 2 and 4 astrocytomas. We identified seventy-four differentially abundant proteins between the grades 2 and 4 astrocytomas by following strict statistical criteria. We shortlisted characteristic proteins from each histological group and thus correlating some with a greater malignancy for this type of tumor. We then used Reactome to list distinctly activated molecular pathways in both groups and corroborated our results with the literature and observations from clinical and surgical practice. We found significant differences in the extracellular matrix organization related to tumor invasion and signaling for cell division, which, together with marked contrasts in energy metabolism, determine factors in the speed of growth and dissemination of these neoplasms. The GABA degradation pathways, more present in the grade 2 group, were consistent with a better prognosis already reported. Other functions such as platelet degranulation, apoptosis, and MAPK approach activation were more related to grade 4 tumors, consistent with a worse prognosis. In all, our results mark a step forward, in the understanding at the molecular level, of these histopathological groups.

## Supporting information

**S1 Table. Complete list of proteins identified in grade 2 and 4 astrocytomas.** The table provides the full list of proteins identified in each tumor grade, including the 748 proteins exclusive to grade 4 astrocytomas, the 174 proteins exclusive to grade 2 astrocytomas, and the 2658 proteins common to both tumor types.
(XLSX)

**S2 Table. Differentially abundant proteins between grade 2 and 4 astrocytomas.** The table contains the list of 74 differentially abundant proteins identified using the TFold module of PatternLab for Proteomics. The abundance of each protein was statistically evaluated based on variable fold change and $q$-value criteria.
(XLSX)

## Acknowledgments

The authors acknowledge the proteomics data generation at Mass Spectrometry Facility RPT02H at Fiocruz—Paraná and thanks Dr. Michel Batista for running the samples.

## Author Contributions

**Conceptualization:** Denildo C. A. Verissimo, Luis A. B. Borba, Paulo C. Carvalho, Juliana de S. da G. Fischer.

**Data curation:** Denildo C. A. Verissimo, Amanda C. Camillo-Andrade, Marlon D. M. Santos, Sergio L. Sprengel, Simone C. Zanine, Luis A. B. Borba.

**Formal analysis:** Denildo C. A. Verissimo, Marlon D. M. Santos, Simone C. Zanine, Paulo C. Carvalho.

**Funding acquisition:** Paulo C. Carvalho.

**Investigation:** Denildo C. A. Verissimo, Amanda C. Camillo-Andrade, Sergio L. Sprengel, Simone C. Zanine, Luis A. B. Borba, Juliana de S. da G. Fischer.

**Methodology:** Denildo C. A. Verissimo, Amanda C. Camillo-Andrade, Marlon D. M. Santos, Simone C. Zanine.

**Project administration:** Denildo C. A. Verissimo, Luis A. B. Borba, Paulo C. Carvalho, Juliana de S. da G. Fischer.

**Resources:** Paulo C. Carvalho.

**Software:** Marlon D. M. Santos, Paulo C. Carvalho.

**Supervision:** Luis A. B. Borba, Juliana de S. da G. Fischer.

**Validation:** Denildo C. A. Verissimo, Paulo C. Carvalho.

**Visualization:** Denildo C. A. Verissimo, Amanda C. Camillo-Andrade, Marlon D. M. Santos, Sergio L. Sprengel, Simone C. Zanine.

**Writing – original draft:** Denildo C. A. Verissimo, Paulo C. Carvalho.

**Writing – review & editing:** Amanda C. Camillo-Andrade, Marlon D. M. Santos, Sergio L. Sprengel, Simone C. Zanine, Luis A. B. Borba.

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
