## [Decision Letter · Decision Letter 0]

18 Apr 2023

PONE-D-23-00607Proteomics reveals differentially regulated pathways when comparing low-versus high-grade astrocytomas.PLOS ONE

Dear Dr. de Saldanha da Gama Fischer,

Thank you for submitting your manuscript to PLOS ONE. After careful consideration, we feel that it has merit but does not fully meet PLOS ONE’s publication criteria as it currently stands. Therefore, we invite you to submit a revised version of the manuscript that addresses the points raised during the review process.

Please respond to all reviewer concerns and extend the disccussion section.

We look forward to receiving your revised manuscript.

Kind regards,

Dragana Nikitovic, Ph.D

Academic Editor

PLOS ONE

Journal Requirements:

https://www.mdpi.com/2072-6694/10/11/441/htm

https://www.frontiersin.org/articles/10.3389/fnmol.2018.00130/full

In your revision ensure you cite all your sources (including your own works), and quote or rephrase any duplicated text outside the methods section. Further consideration is dependent on these concerns being addressed.

JSGF received a scholarship from covenant APU / PUCPR 28/2021 (Fundação Araucária) 

Carlos Chagas. Institute (Fiocruz Paraná) for reagents and infraestructure support.

The authors also acknowledge Carlos Chagas Institute (Fiocruz Paraná) and covenant APU / PUCPR 28/2021 (Fundação Araucária) for financial support. 

JSGF received a scholarship from covenant APU / PUCPR 28/2021 (Fundação Araucária) 

Carlos Chagas. Institute (Fiocruz Paraná) for reagents and infraestructure support.

6. Thank you for stating the following in your Competing Interests section:  

NO authors have competing interests

Additional Editor Comments:

Please respond to reviewers' comments.

Reviewers' comments:

Reviewer's Responses to Questions

**Comments to the Author**

1. Is the manuscript technically sound, and do the data support the conclusions?

Reviewer #1: Yes

Reviewer #2: Yes

2. Has the statistical analysis been performed appropriately and rigorously? 

Reviewer #1: Yes

Reviewer #2: Yes

3. Have the authors made all data underlying the findings in their manuscript fully available?

Reviewer #1: No

Reviewer #2: Yes

4. Is the manuscript presented in an intelligible fashion and written in standard English?

Reviewer #1: Yes

Reviewer #2: Yes

5. Review Comments to the Author

Reviewer #1: The research demonstrated that provide an important survey of molecular pathways involved in glioma pathogenesis for these histopathological groups. However there are some points to improve.

1.The current WHO guidelines for gliomas have been updated to WHO CNS5, and this article uses a simple classification of astrocytomas into lower and higher grades for comparison, which may not be appropriate.

2.The statistical methods used in this paper were relatively simple and no further validation of the results was performed.

3.This paper does not present new findings and is relatively low in novelty.

4.The discussion in this study is not deep enough, and it is suggested that it be discussed in depth.

Reviewer #2: The authors of the manuscript PONE-D-23-00607 titled “Proteomics reveals differentially regulated pathways when comparing low-versus high-grade astrocytomas” have answered to all the reviewer’s points. The manuscript is well written and they present and discuss their results nicely. I therefore agree to the publication of the manuscript.

6. PLOS authors have the option to publish the peer review history of their article (what does this mean?). If published, this will include your full peer review and any attached files.

Reviewer #1: No

Reviewer #2: No

---

## [Author Response · Author response to Decision Letter 0]

13 May 2023

Journal Requirements:

We made a minor adjustment to the corresponding author’s indication to abide to the templates above.

https://www.mdpi.com/2072-6694/10/11/441/htm

https://www.frontiersin.org/articles/10.3389/fnmol.2018.00130/full

We appreciate the reviewer's attention to detail in identifying the overlapping text with previous publications. We rephrased and expanded the relevant sections to ensure uniqueness and to differentiate our work from the cited references. We trust that these changes will address the concern raised and improve the clarity of our manuscript.

In regards to reference (60, i.e., https://www.frontiersin.org/articles/10.3389/fnmol.2018.00130/full), we believe the minor overlapping text was:

“Further investigations also revealed that fibronectin promoted cell growth, as demonstrated by the elevation of Ki-67, with activation of p-ERK1/2 and cyclin D1. Moreover, fibronectin suppressed p53-mediated apoptosis and upregulated P-glycoprotein expression.”

Of which we re-worded to

“Additional studies demonstrated that fibronectin facilitated cell proliferation, as evidenced by increased Ki-67 levels, along with the activation of phosphorylated ERK1/2 and cyclin D1. Furthermore, fibronectin was found to inhibit apoptosis mediated by the p53 pathway and enhance the expression of P-glycoprotein.”

And for reference (75, i.e., https://www.mdpi.com/2072-6694/10/11/441/htm)

We believe the minor overlapping section was:

“Platelets can protect circulating tumor cells against the immune system, favor pro-survival signals, induce invasive properties and transfer adhesive molecules that interact with the endothelium participating in early metastatic niches (75).”

Of which we re-worded to

“Platelets possess multifaceted roles in cancer progression, including the ability to safeguard circulating tumor cells from immune surveillance, fostering an environment that supports cell survival. Additionally, they can stimulate invasive characteristics in tumor cells, enhancing their metastatic potential. Platelets also facilitate the transfer of adhesive molecules, which play a crucial part in establishing interactions between tumor cells and the endothelial lining of blood vessels. This interaction aids in the formation of early metastatic niches, further driving the spread of cancer throughout the body (75).”

In your revision ensure you cite all your sources (including your own works), and quote or rephrase any duplicated text outside the methods section. Further consideration is dependent on these concerns being addressed.

This, notwithstanding, we have taken the opportunity to review and rephrase the Materials and Methods section to provide a refreshed perspective on the experimental protocol with non-overlapping text of our previous publications. Although it was not strictly necessary, we believe that these revisions have enhanced the clarity and comprehensibility of our methods. We hope that the editor finds the updated section satisfactory and that it contributes to the overall quality of our manuscript. We remain committed to addressing any further concerns and are grateful for the opportunity to continually improve our work.

We appreciate your attention to the details regarding participant consent. We have modified the manuscript to better reflect the consent process followed in our study as per request. We have specified that informed written consent was obtained from all subjects and/or their legal guardian(s) and that all data were fully anonymized before analysis to ensure participant privacy. Please find the revised statement in the manuscript for your reference.

Before:

“All methods were performed in accordance with the relevant guidelines and regulations of Brazil. This study was approved by the Ethics Committee of Oswaldo Cruz Foundation and the Federal University of Clinical Hospital of Parana under the numbers CAAE: 63056316.8.0000.5248 and 63056316.8.3001.0096, respectively. Informed consent was obtained from all subjects and/or their legal guardian(s).”

Now:

“All methods were performed in accordance with the relevant guidelines and regulations of Brazil. This study was approved by the Ethics Committee of Oswaldo Cruz Foundation and the Federal University of Clinical Hospital of Parana under the numbers CAAE: 63056316.8.0000.5248 and 63056316.8.3001.0096, respectively. Informed written consent was obtained from all subjects and/or their legal guardian(s) before participating in the study. All participants were provided with detailed information about the study, its objectives, and the use of their data for research purposes. No minors participated in this study. All data were fully anonymized before analysis to ensure participant privacy.”

JSGF received a scholarship from covenant APU / PUCPR 28/2021 (Fundação Araucária) 

Carlos Chagas. Institute (Fiocruz Paraná) for reagents and infrastructure support.

Thank you for pointing this out. We have amended as requested in the manuscript and reflected below for your convenience:

JSGF received a scholarship from covenant APU / PUCPR 28/2021 (Fundação Araucária),

Carlos Chagas Institute (Fiocruz Paraná) for reagents and infrastructure support. We also acknowledge the grant PEP-ICC-008-FIO-21 from Fiocruz. The funders had no role in study design, data collection and analysis, decision to publish, or preparation of the manuscript.

The authors also acknowledge Carlos Chagas Institute (Fiocruz Paraná) and covenant APU / PUCPR 28/2021 (Fundação Araucária) for financial support. 

JSGF received a scholarship from covenant APU / PUCPR 28/2021 (Fundação Araucária) 

Carlos Chagas. Institute (Fiocruz Paraná) for reagents and infrastructure support.

Thank you for pointing this out. We have removed the referred information from the acknowledgements and placed it in our cover letter so you can change it on our behalf. The full funding statement now reads:

JSGF received a scholarship from covenant APU / PUCPR 28/2021 (Fundação Araucária); 

Carlos Chagas Institute (Fiocruz Paraná) for reagents and infrastructure support. We also acknowledge the grant PEP-ICC-008-FIO-21 from Fiocruz. The funders had no role in study design, data collection and analysis, decision to publish, or preparation of the manuscript.

6. Thank you for stating the following in your Competing Interests section: 

NO authors have competing interests

We changed the cover letter as requested.

We have included the captions for the Supporting Information files at the end of the manuscript as requested. Please find them reflected below for your convenience:

S1 Table. Complete list of proteins identified in grades 2 and 4 astrocytomas. The table provides the full list of proteins identified in each tumor grade, including the 748 proteins exclusive to grade 4 astrocytomas, the 174 proteins exclusive to grade 2 astrocytomas, and the 2658 proteins common to both tumor types.

S2 Table. Differentially abundant proteins between grades 2 and 4 astrocytomas. The table contains the list of 74 differentially abundant proteins identified using the TFold module of PatternLab for Proteomics. The abundance of each protein was statistically evaluated based on variable fold change and q-value criteria.

To adhere to the journal’s standards, we now refer to these tables as S1 Table, S2 Table, and have renamed the files to S1_Table.xls and S2_Table.xls, respectively.

Additional Editor Comments:

Please respond to reviewers' comments.

Dear Dragana Nikitovic, Ph.D,

Thank you for considering our manuscript. We appreciate the time and effort that you and the reviewers have put into evaluating our work.

We have carefully addressed the comments and suggestions provided by the reviewers and have revised the manuscript accordingly. Below, we reflect the points raised by the reviewers inline with a reply for each suggestion.

Reviewers' comments:

Reviewer's Responses to Questions

Comments to the Author

1. Is the manuscript technically sound, and do the data support the conclusions?

Reviewer #1: Yes

Reviewer #2: Yes

2. Has the statistical analysis been performed appropriately and rigorously?

Reviewer #1: Yes

Reviewer #2: Yes

3. Have the authors made all data underlying the findings in their manuscript fully available?

Reviewer #1: No

Reviewer #2: Yes

Reply: We would like to clarify Reviewer #1's concern regarding data availability. In our manuscript, we have made all the raw mass spectrometry data fully available via the PRIDE repository, as indicated in the manuscript, along with our search results. We have used open-source software and provided tables with patient data and differentially abundant proteins in the manuscript itself. We believe the Reviewer #1 marked "No" for data availability as we forgot to provide username and password for the reviewer to access our raw files before publication; this is now provided below. We now believe to have complied with the PLOS Data policy by making all the underlying data fully accessible without restriction.

“Availability of data 

The mass spectrometry data was deposited to the ProteomeXchange Consortium via the PRIDE (83) partner repository with the dataset identifier PXD033782.”

The reviewer can access this dataset before publication using the following information:

 Username: reviewer_pxd033782@ebi.ac.uk

 Password: t1sFPBkf

4. Is the manuscript presented in an intelligible fashion and written in standard English?

Reviewer #1: Yes

Reviewer #2: Yes

5. Review Comments to the Author

Reviewer #1: The research demonstrated that provide an important survey of molecular pathways involved in glioma pathogenesis for these histopathological groups. However there are some points to improve.

1.The current WHO guidelines for gliomas have been updated to WHO CNS5, and this article uses a simple classification of astrocytomas into lower and higher grades for comparison, which may not be appropriate.

We appreciate the reviewer's recommendation to include the WHO CNS5 guidelines in our study. We acknowledge that the updated guidelines, incorporating molecular markers such as IDH mutation status and 1p/19q codeletion, offer a more accurate diagnosis and classification of astrocytomas. However, due to limited resources and the unavailability of these tests in Brazilian public services, these additional analyses are not provided. Nonetheless, this fact is also foreseen in the WHO CNS5 classification with the suffix NOS (not otherwise specified).

We understand that molecular advancements have provided valuable insights into the prognosis of tumors previously considered as grade 2 (diffuse astrocytomas) but may exhibit more malignant behavior. For instance, the presence of IDH mutations confers a better prognosis, while the rare IDH wild-type tumors of this type indicate an unfavorable prognosis, even recommending treatment approaches similar to glioblastomas.

To address this issue, we have changed the terms "low-grade" and "high-grade" to "grade 2 astrocytomas" and "grade 4 astrocytomas" in our work, based mainly on histopathological features and, according to WHO CNS5, the patients' natural history. By doing so, we maintain an appropriate comparison between these subgroups while acknowledging the limitations of more specific molecular subclassifications.

This notwithstanding, we updated the legend to Table 1 and included the following paragraph in our discussion to acknowledge this limitation:

"In our study, we acknowledge the limitation of not incorporating all the WHO CNS5 guidelines for astrocytoma classification due to the unavailability of certain molecular marker tests, such as IDH mutation status and 1p/19q codeletion, in vast majority of Brazilian public services. This fact is also foreseen in the 5 WHO classification with the suffix NOS (not otherwise specified). Despite these limitations, our research provides valuable information on the proteomic differences between grade 2 and grade 4 astrocytomas based on their histopathological features and patients' natural history. We recognize that the integration of molecular markers in astrocytoma classification could offer a more accurate diagnosis and improved understanding of tumor biology, leading to better patient stratification and tailored treatment approaches. We encourage future research to build upon our findings and incorporate the WHO CNS5 guidelines for a more comprehensive analysis of astrocytomas and their underlying molecular pathways."

We hope this addresses the reviewer's concern and believe that our study still provides valuable insights into the proteomic differences between grade 2 and grade 4 astrocytomas based on their histopathological features and patients' natural history.

2.The statistical methods used in this paper were relatively simple and no further validation of the results was performed.

We acknowledge the concern regarding the lack of further validation in our study. The process of obtaining the samples for this research was quite challenging and time-consuming, taking approximately one and a half years to accomplish. Additionally, the resources and availability of specific reagents for validation are limited in our setting. Despite these challenges, we rigorously followed a well-established bioinformatics protocol for analyzing proteomic data, following all steps from our last year’s Nature Protocol on analyzing shotgun proteomic data (https://pubmed.ncbi.nlm.nih.gov/35411045/). This approach has proven effective in identifying differentially abundant proteins and elucidating molecular pathways. Moreover, our stringent criteria for protein selection, along with the corroboration of our results with existing literature and clinical observations, contribute to the validity of our findings. While we understand the importance of further validation, we believe that our current results provide valuable insights into the molecular differences between astrocytomas grade 2 and 4. 

3.This paper does not present new findings and is relatively low in novelty.

While it is true that some of the molecular pathways we identified have been previously reported in the context of astrocytomas, our study provides a comprehensive comparison of proteomes from astrocytomas grade 2 and 4 using a unique set of samples collected over a significant period of time. The identification of 74 differentially abundant proteins, many of which are in accordance with the literature (and thus indirectly validated, at least to some extent), as well as the disclosure of several novel proteins, adds valuable information to the existing body of research. Our work not only confirms previously reported findings but also serves as a foundation for future research to validate and explore the roles of these newly identified proteins in astrocytoma biology. Ultimately, this could lead to a better understanding of the underlying molecular mechanisms and improved patient management and treatment strategies.

4.The discussion in this study is not deep enough, and it is suggested that it be discussed in depth.

We included additional discussion as reflected below for your convenience and note that our discussion is already over 13 word-pages long:

Considering the results taken together, the identified alterations in extracellular matrix organization, cell division signaling, and energy metabolism offer valuable insights into the mechanisms contributing to the distinct characteristics of these histopathological groups. For example, our study highlights the importance of GABA degradation pathways in grade 2 astrocytomas and their implications for understanding neurotransmitter signaling in glioma pathogenesis. Investigating the relationship between these pathways and tumor cell behavior may help elucidate why grade 2 astrocytomas are associated with a better prognosis. In contrast, the increased presence of platelet degranulation, apoptosis, and MAPK pathway activation in grade 4 astrocytomas suggests a more aggressive tumor phenotype. Understanding how these processes influence tumor growth and invasion could facilitate the identification of therapeutic targets or intervention strategies for grade 4 astrocytomas.

Our study also emphasized the significance of dysregulated energy metabolism in astrocytoma pathogenesis by pinpointing differences in energy metabolism pathways between grades 2 and 4 astrocytomas, which may contribute to their distinct growth rates and invasiveness. Investigating the specific metabolic alterations and their impact on tumor cell proliferation, survival, and migration could reveal novel therapeutic targets to hinder tumor progression.

In all, exploring the interplay between differentially abundant proteins and molecular pathways may provide a more holistic understanding of the complex cellular processes driving astrocytoma development and progression. Further studies examining these interactions to shortlist regulatory nodes are key to better therapeutic targeting to disrupt tumor growth and spread. Future studies should focus on these relationships and the underlying mechanisms, paving the way for more effective and personalized treatment strategies for patients with astrocytomas.

Reviewer #2: The authors of the manuscript PONE-D-23-00607 titled “Proteomics reveals differentially regulated pathways when comparing low-versus high-grade astrocytomas” have answered to all the reviewer’s points. The manuscript is well written and they present and discuss their results nicely. I therefore agree to the publication of the manuscript.

Thank you for taking the time to review our manuscript. We appreciate your evaluation and are glad that you did not find any concerns in our work. We hope our research contributes to the current understanding of the molecular pathways involved in grades 2 and 4 astrocytomas and serves as a valuable resource for future studies in this field.

---

## [Decision Letter · Decision Letter 1]

2 Aug 2023

Proteomics reveals differentially regulated pathways when comparing grade 2 and 4 astrocytomas.

PONE-D-23-00607R1

Dear Dr. de Saldanha da Gama Fischer,

We’re pleased to inform you that your manuscript has been judged scientifically suitable for publication and will be formally accepted for publication once it meets all outstanding technical requirements.

Kind regards,

Dragana Nikitovic, Ph.D

Academic Editor

PLOS ONE

Additional Editor Comments (optional):

Reviewers' comments:

Reviewer's Responses to Questions

**Comments to the Author**

1. If the authors have adequately addressed your comments raised in a previous round of review and you feel that this manuscript is now acceptable for publication, you may indicate that here to bypass the “Comments to the Author” section, enter your conflict of interest statement in the “Confidential to Editor” section, and submit your "Accept" recommendation.

Reviewer #1: (No Response)

Reviewer #2: All comments have been addressed

2. Is the manuscript technically sound, and do the data support the conclusions?

Reviewer #1: (No Response)

Reviewer #2: Yes

3. Has the statistical analysis been performed appropriately and rigorously? 

Reviewer #1: (No Response)

Reviewer #2: Yes

4. Have the authors made all data underlying the findings in their manuscript fully available?

Reviewer #1: (No Response)

Reviewer #2: Yes

5. Is the manuscript presented in an intelligible fashion and written in standard English?

Reviewer #1: (No Response)

Reviewer #2: Yes

6. Review Comments to the Author

Reviewer #1: The author has provided some explanations in response to the concerns raised, but it seems that the fundamental issue of limited novel results in this study has not been fully addressed. It is important to recognize that the research may have produced fewer new findings than expected or desired.

Reviewer #2: The authors of the manuscript titled "Proteomics reveals differentially regulated pathways when comparing grade 2 and 4

astrocytomas." have answered all the comments necessary.

7. PLOS authors have the option to publish the peer review history of their article (what does this mean?). If published, this will include your full peer review and any attached files.

Reviewer #1: No

Reviewer #2: No

---

## [Editor Report · Acceptance letter]

15 Aug 2023

PONE-D-23-00607R1 

proteomics reveals differentially regulated pathways when comparing grade 2 and 4 astrocytomas. 

Dear Dr. da G. Fischer:

I'm pleased to inform you that your manuscript has been deemed suitable for publication in PLOS ONE. Congratulations! Your manuscript is now with our production department. 

Kind regards, 

on behalf of

Dr. Dragana Nikitovic 

Academic Editor

PLOS ONE